# Femtosecond Laser Fabrication of Microporous Membranes for Biological Applications

**DOI:** 10.3390/mi13091371

**Published:** 2022-08-23

**Authors:** Annalisa Volpe, Filippo Maria Conte Capodacqua, Valeria Garzarelli, Elisabetta Primiceri, Maria Serena Chiriacò, Caterina Gaudiuso, Francesco Ferrara, Antonio Ancona

**Affiliations:** 1Physics Department, Università degli Studi di Bari & Politecnico di Bari, Via Orabona 4, 7016 Bari, Italy; 2Institute for Photonics and Nanotechnologies (IFN), National Research Council, Via Amendola 173, 70126 Bari, Italy; 3CNR NANOTEC—Institute of Nanotechnology, Via per Monteroni, 73100 Lecce, Italy; 4Department of Mathematics & Physics E. de Giorgi, University of Salento, Via Arnesano, 73100 Lecce, Italy

**Keywords:** fs-laser, micromachining, laser drilling, membrane, cell manipulation

## Abstract

The possibility of fabricating micrometric pore size membranes is gaining great interest in many applications, from studying cell signaling, to filtration. Currently, many technologies are reported to fabricate such microsystems, the choice of which depends strictly on the substrate material and on the final application. Here, we demonstrate the capability with a single femtosecond laser source and experimental setup to fabricate micromembranes both on polymeric and multilayer metallic substrate, without the need for moulds, mask, and complex facilities. In particular, the flexibility of laser drilling was exploited to obtain microfilters with pore size of 8 and 18 µm in diameter, on metallic and polymeric substrate, respectively, and controlled distribution. For evaluating the possibility to use such laser-fabricated membranes into biological assay, their biocompatibility has been investigated. To this aim, as a proof of concept, we tested the two materials into viability tests. The culture of mammalian cells on these microfabricated membranes were studied showing their compatibility with cells.

## 1. Introduction

The capability of fabricating 3D microstructures with arbitrary shape is gaining interest in many biomedical fields, from cell manipulation [1] to stent manufacture [2]. In particular, the fabrication of filters consisting of pores of controllable micrometric size and distribution is opening the possibility of direct interaction among particles and/or cells.

The targeted pores size of the membranes depends on the application for which they would be used [3], i.e., for studying cell-cell signaling [4], cell migration [5], or filtration [6]. Membranes with pores of around 10 μm in diameter are used to investigate cell migration events, where the membranes facets can be functionalized with an extracellular matrix component (collagen, laminin or fibronectin) in order to evaluate extracellular conditions [7], or to test drugs to contrast metastatic processes [8,9]. Whereas, for diameters smaller than 1 μm, cell migration through the membrane pores is usually blocked [4] and two cell populations can be seeded and grown separately, interacting just through soluble factors while remaining unmixed. Lack of mixing is important for studies in which each population needs to be analyzed separately, for example to assay cell-specific changes in gene or protein expression.

Either polymer, silicon, or glass [10] are reported in literature as materials used for the fabrication of membranes with micrometric pores size.

Once identified the final application and according to the material selected as the substrate, several techniques can be employed for the micro-pores membrane fabrication.

Polymeric membranes are the most popular due to the low cost, biocompatibility and relative wide choice of manufacturing techniques. Nowadays, commercial polymeric membranes are mainly produced by track etching (TE) technique [11,12]. Track etching is easily scalable and allows to produce pore with dimensions ranging from 0.01 µm to 30 µm on many polymers, e.g., PET, PC, PP, and PI [11]. A polycarbonate track-etch-type membrane with 8 µm-diameter cylindrical pores was reported by Vona et al. [13]. The main disadvantages of this technology are that it requires a heavy-ion accelerator and, due to the random pores’ placement, low porosity is achieved in order to prevent pores overlapping.

Another common strategy to produce polymeric micro-membranes is the phase separation method [3]. Exploiting this technique, and tuning the fabrication parameters, Zhao et al. [14] prepared a microporous silicone rubber membrane with pore in a wide range of diameters (from 3 to 41 µm). The process is inexpensive and simple, however, the distribution of the pores cannot be controlled, as well as their density. Moreover, the section of the pores is blind and, thus, more indicated for application in biomedical field such as artificial skin and scaffold for tissue engineering.

To overcome the limitations of track etching and phase separation methods, Kim et al. [15] developed a photolithography-based microfabrication technique to achieve PET membranes with up to 40% porosity, and less than 5% variation in pore size. Their membrane is 1 µm-thick, with pores diameter of 0.9 μm. This approach can achieve similar pore sizes as the track etching method, but here the pores can be placed precisely rather than randomly, thus allowing high pore density without overlapping.

Membranes have been also fabricated by soft lithography in conjunction with chemical etching in polydimethylsiloxane (PDMS) [16] with area sizes of the order of 1 cm^2^, which is comparable to commercial cell culture membranes and holds adequate cell numbers for standard protein and nucleic acid quantification assays. The authors achieved a 10-µm-thick PDMS membrane containing an array of through-holes with an effective diameter of 10 µm. Despite the possibility to precisely arrange the pores, e.g., in hexagonal and rectangular patterns, the aforementioned technique needs a mould and no-green chemical reagents.

A vacuum assisted UV micro-moulding (VAUM) process has been proposed [17] for the fabrication of freestanding polymer membranes based on a UV-curable methacrylate polymer (MD 700). The proposed technique allows to fabricate low cost, robust, large-area membranes up to 9 × 9 cm^2^ with pore sizes from 8 to 20 μm in diameter, 20 to 100 μm in thickness, high aspect ratio (the ratio of the thickness of the polymer over the diameter of the hole is up to 15:1), high porosity, and a wide variety of geometrical characteristics. Unfortunately, a different master mould needs to be fabricated for each membrane design, thus this technique has limited flexibility.

Although most of the literature concerns the manufacturing of polymeric membranes, a few studies deal with other materials, i.e., ceramics and metals.

Electron beam lithography has been utilized to produce silicon-nitride membranes with pore sizes down to 0.3 μm and 0.5 μm in thickness [18]. While these dimensions are excellent, the cost of semiconductor processing is high and electron beam lithography is a serial writing process, making the production of large-area membranes impractical (<1 mm^2^ in the cited work).

A 30 μm-thick silicon membrane with a uniform pore diameter (10 μm) fabricated using a deep reactive ion etching (DRIE) process is also reported. The authors presented a lab-on-a-chip device for highly efficient and rapid separation of circulating tumor cells (CTCs) from whole blood samples [19]. The device utilizes a microfabricated silicon microsieve with a densely packed pore array (105 pores per device) to rapidly separate tumour cells from whole blood, utilizing the size and deformability differences between the CTCs and normal blood cells. The DRIE process allowed producing micropores with uniform and precise depth. However, it is a multi-step process (the authors listed 5 steps), not easily reconfigurable due to the need of a mask.

While there are many studies on membranes made from polymers and silicon, there are limited applications of metal membranes, despite their unique advantages over all other membrane materials, including excellent mechanical strength and chemical resistance [20]. Porous metal membranes with micrometric pore size (1–100 µm) are principally made by sintering or foaming methods. These are, cheap and easily scalable manufacturing techniques, but which do not allow any control over pores distribution and density. Actually, metallic porous membranes are mainly used for the filtration of liquids [21] to remove the solid particles, but metallic membranes for bacteria retention have also been reported in drug delivery devices, and in medical instrumentation, e.g., as bacteria filters [22].

Femtosecond (fs) laser technology has been demonstrated to be a flexible and mould-free technique to fabricate devices with micrometric precision [23] in a plethora of applications from microfluidics [24] to surface texturing [25]. Fs-laser enables energy deposition at a shorter timescale than the electron–phonon coupling processes, which allows the material to be removed by laser ablation from the irradiated area with negligible thermal damage to the surrounding substrate. In this way, it is possible to create micromilled features with high precision and resolution [26] almost completely free from thermally induced defects like surface or subsurface cracks, residual stresses, resolidified melting and burrs. This non-contact method does not pose any constraint on the material choice. Indeed, laser fabrication of microdevices have been reported, e.g., on polymers [27], ceramics [28], metals [25], etc. The large plethora of materials machinable with lasers opens the way to the fabrication of membranes by direct fs-laser ablation, which can meet the need for customization in biological assays. Indeed, Yalikin et al. [29] introduce 4-µm- thick glass filter. Membrane with through holes with a diameter of approximately 5 µm (e.g., aspect ratio ~1) were fabricated to be introduced in an all-glass microfluidic cell culturing device without circulation flow [30]. The function of this all-glass culturing device was confirmed by culturing HeLa, fibroblast and ES cells.

In this work, we demonstrate the potential of fs-laser drilling to fabricate freestanding microfilters both on polymeric and metallic substrates, cellulose acetate (CA) and Copper-Kapton-Copper (CKC) multilayer, respectively. In a single step, we obtain membranes without restrictions on the substrate material chosen for each specific application and with higher process efficiency than previous fs-laser based fabrication works. The flexibility of laser drilling allows to obtain micro-pores matrix with controlled distribution that can be easily changed according to the biological study to be performed. As a proof of concept, we tested the micro-fabricated membranes into viability tests, demonstrating their compatibility with mammalian cells.

## 2. Materials and Methods

### 2.1. Materials

Two different substrates were used for the fabrication of the membranes: (i) a multi-layer material Copper-Kapton-Copper (CKC), composed by a 50-µm-thick layer of Kapton, sandwiched between two 5-µm-thick chemical vapour-deposited layers of copper, and (ii) a 150 µm-thick cellulose acetate (CA) monolayer. The choice of these two substrates was made based on their different characteristics and to demonstrate the flexibility offered by the ultrafast laser fabrication technology to machine different materials with micrometric precision. CA is a biocompatible [31], low-cost polymer, exceptionally transparent. The CKC multilayer has excellent physical, chemical, and electrical properties. Despite the cytotoxicity of copper, this multilayer substrate is a good candidate to carry out electrical measurement directly on a chip.

### 2.2. Micro-Membrane Design and Fabrication

The sketch of the setup exploited for the microporous membranes fabrication is shown in Figure 1.

A compact Yb:KGW femtosecond laser system (PHAROS from Light Conversion) based on the chirped pulse amplification technique, was used delivering an almost diffraction limited beam (M^2^ ∼1.3) with ultrashort pulses of 200 fs duration, central wavelength *λ* 1030 nm, maximum pulse energy of 1.5 mJ, and variable repetition rate from single pulse to 1 MHz. The circular polarized laser beam was focused by a 20× microscope objective with numerical apertures NA of 0.50 and focal length *f* 8 mm (Newport Corporation, Irvine, CA, USA). Assuming that the laser was not a theoretical Gaussian beam, the calculated diameter of the focused laser spot was 4.7 µm and was calculated by 4*λfM*^2^/*πd*, with *d* laser beam diameter at lens [32]. This value limits the minimum pore diameter that can be achieved with the employed objective lens. The calculated Rayleigh range was about 67 µm. The objective was mounted on a computer controlled motorized air bearing stage (Aerotech, ANT130 LZS, Pittsburgh, PA, USA) enabling to precisely position the beam focus over the work-pieces. The substrates were fixed on an air bearing XY motorized translation table (Aerotech, ABL1500 LZS, Pittsburgh, PA, USA) with sub-micrometric positioning resolution. As the samples held on an automated stage was moved, the through holes were fabricated one by one by laser percussion drilling.

In Table 1, the process parameters are summarized for both the substrates.

These working parameters are the summary of a preliminary study aimed at finding the best combination of parameters that guarantee a good pore quality and an efficient process for both materials. The study is out of the scope of this article. However, we can briefly assert that it plays a key role the laser energy rate given to the material [33]. At high frequency, heat accumulation can cause the melting of the CA plastic substrate, so on this material we work at about 100 times lower frequency than on CKC, in which the copper layers guarantee a good heat dissipation [34]. Moreover, a higher fluence is required for CA, in order to trigger nonlinear absorption processes in the dielectric material [35]. Consequently, we found a different number of shots per hole, namely 2500 and 30 laser shots for CKC and CA, respectively, but more energetic for the pure polymer and at a lower repetition rate.

Exploiting the flexibility of the laser technology, membranes with three different distributions of the pores (Figure 2b–d), have been fabricated on both types of substrates. The membranes were laser-cut with a circular shape of diameter 18 mm (Figure 2a), in order to be easily housed in a cylindrical chamber designed for biological tests. Considering the design of the chamber, the effective contact area of the membrane with the cells was a circle with a diameter of 14 mm. The desired pore diameter size for the target cell migration application was comprised between 1 μm and 20 μm.

After laser irradiation, the samples were sonicated for ten minutes in distilled water and the laser-induced modifications were inspected through optical microscopy, to evaluate the entry and exit diameters of the pores.

### 2.3. Cell Culture, Sample Preparation and Cell Adhesion Experiments

PC-3 human Cell Line (Caucasian prostate adenocarcinoma, 90112714) was cultured in Roswell Park Memorial Institute (RPMI) 1640 Medium with L-glutamine and sodium bicarbonate, adding Fetal Bovine Serum 10% (FBS-F7524), Penicillin-Streptomycin 1% (P0781) and sodium pyruvate 1%. Cells and reagents were purchased from Sigma Aldrich Company (Sigma Aldrich, St. Louis, MO, USA). PC-3 cells were seeded in 24 well culture plates (50.000 cells/500 µL/well) and incubated at 37 °C, 5% CO_2_ incubator. To avoid copper cytotoxic intrinsic effect [36,37] and improve biocompatibility, both faces of the CKC ablated membranes were coated with a nanometric gold layer (10 nm) by thermal evaporation. In Figure 3 one side of the CKC membrane is reported.

CA and gold coated-CKC membranes of size 5 mm × 5 mm were incubated overnight in FBS or in a solution 10% of Fibronectin (F4759-Sigma Aldrich) in PBS. Control samples were not functionalized.

Cells seeded in the 24 well plates were then kept in contact with the functionalized fragments and control fragments (triple replicates) and left for 24 h. A methyl thiazolytetrazolium (MTT) assay was then performed. MTT test allows to evaluate the cell viability. It is a colorimetric assay based on the enzymatic activity of succinate dehydrogenase that converts yellow and soluble MTT tetrazolium salts into purple and insoluble formazan crystals salts 31. 500 µL of Thiazolyl Blue Tetrazolium Bromide solution (MTT-M2128, Sigma Aldrich) from the Stock solution (5 mg/mL) was diluted 1:10 in culture medium and added to each well. The plates were incubated at 37 °C, 5% CO_2_ in dark environment for 3 h. The medium was then removed, and Formazan crystals formed by cells were dissolved using 1 mL of isopropanol for each well. The absorbance was read at 570 nm on CLARIOstarPlus (BMG Labtech) multiwell plate reader.

PC-3 cells were seeded on the surface of Gold-coated CKC and CA membranes to check their ability to adhere to these two substrates and demonstrate that these membranes can be used and incorporated in microsystems for complex cell investigation such as migration or co-culture assays.

In both cases, according to previous viability tests, membranes were functionalized overnight with FBS before being kept in contact with cells. Then cells were fixed with paraformalheide 3.7% and stained with DAPI (Sigma Aldrich, St. Louis, MO, USA).

Not adhering cells were removed from the surface during the washing steps while the adhering ones could be observed in brightfield and fluorescence.

A proof-of-concept migration experiment was performed on CKC gold-coated membrane. To this aim a 35 mm Petri-dish was modified with a PDMS ring allowing to lift up the membrane from the bottom of the plastic dish and to identify migrated cells on the reverse side of the membrane. We seeded PC3 cells on the tapered surface of CKC membranes keeping them in standard cell culturing conditions for 24 h. At the end of the experiment, we disassembled the set up and the membrane was observed on the reverse side by fluorescence microscope.

## 3. Results

### 3.1. Fs-Laser Fabrication of Multilayers and Polymeric Membranes

#### 3.1.1. Laser Fabrication of CKC Membrane

Due to the Gaussian profile of the laser beam, the pores were tapered at an angle θ of about 10° [38], as shown in Figure 4. Consequently, the hole diameters on the top and bottom layer are different. The conical shape is typical of laser drilled holes and is one of the inherent manufacturing problems associated with laser percussion drilling [39].

In Figure 5 the bottom surfaces of Cu-Kapton-Cu (CKC) multilayer membranes are shown. In the inset of Figure 4, the pores distribution is highlighted at two different magnifications.

The minimum dimension obtained for the exit diameter of the pores was 8 µm. In Table 2 the pores number for each configuration, with the corresponding pore-pore distance and pore exit diameter are summarized. The reported density of the pores takes into account the entire area of the membrane (about 154 mm^2^).

The fabrication time depends on the number of holes, but, in all cases, is in the range of few tens of minutes per filter. Taking into account the high number of pores, our process times are about 100 times more efficient than fs-laser technology previously developed for the fabrication of 4-μm-thick glass filter [29].

#### 3.1.2. Laser Fabrication of CA Membrane

In case of CA, a minimum pore exit diameter of about 18 µm was obtained, confirming the taper angle of about 10°, similarly to what observed for the CKC substrate. In Figure 6a magnification of the inlet and outlet facet of a pore is shown.

The quality of the pores on the CA appears worse than those on the doubled side metal coated Kapton film. This difference can be attributed to the base material which, being a dielectric, dissipates heat worse than copper causing unwanted melting at the edges of the hole. Moreover, the thickness of the CA layer is about 3 times the thickness of the CKC. Being the focal position fixed during each pore fabrication, during the drilling the focus is lost, and the process loses efficiency and quality.

In order to avoid any overlap between adjacent pores, the distance between consecutive holes was increased to 200 µm, but maintaining the configurations indicated in Figure 3, so that the overall number of pores for the CA membranes was lower compared to the CKC ones (see Table 3).

In some applications, optical transparency of the membrane is required to allow live-imaging of the cell culture [10]. Therefore, taking advantage of the flexibility of fs-laser fabrication technique, we produced other membranes with a larger pore separation of 400 µm but preserving the pore number reported in Table 3. In this way, the same number of pores was distributed over a larger area and the overall transparency was improved. In Figure 7 the configuration 3 is shown both for the pore distance (a) 200 µm and (b) 400 µm.

### 3.2. Biological Assays

#### 3.2.1. Biocompatibility Tests

As described in Materials and Methods section, MTT assay was performed to evaluate the potential cytotoxicity of materials used for membrane microfabrication. Observing the graph in Figure 8a, it is possible to highlight that after 24 h of cells incubation with Gold-coated CKC membranes, there is a significant reduction (*p* value < 0.05) of more than 50% in PC-3 cell viability compared to control (cells without any fragment).

In addition, in both cases (Multi-layered and CA substrates) the fragments incubation with FBS shows a cell viability increase in comparison to fibronectin and absence of functionalization.

In particular, for gold-coated CKC the reduction in viability is of 51.3% for fibronectin, of 34.9% for FBS and of 54.7% in absence of functionalization.

In parallel, in Figure 8b the CA fragments show less cytotoxicity than the CKC multilayer. The trend of cell viability with two post-functionalized materials in the MTT assay is the same, because the number of dead cells increases. Death increases when non-functionalized CA (77.8% of cells alive) is used compared with Fibronectin (82.4% of cells alive) and FBS (91% of cells alive).

These results clearly demonstrate that pre-incubating the metal membrane and CA membrane in serum, prior to keep substrates in contact with cells, improves the biocompatibility of the two materials tested. To sum up, it is possible to prove that in every condition tested (Fibronectin, FBS and no functionalization), the cells viability is more pronounced on the CA membrane than on the CKC one.

#### 3.2.2. Cell Adhesion Experiments

Based on MTT assay results, we decided to test the adhesion of PC-3 cell lines on the surface of the microfabricated membranes (obtained both on multi-layered and CA substrates). The top and bottom surfaces of Figure 4, once the membrane has been reversed, represents the migration and seeding side of the biological assay, respectively.

AXIO Zoom.V16 microscopy images in bright-field and fluorescence (Figure 9a,b) on the seeding side of the metal membrane display PC-3 cells adhering on region near the pores of the membrane. To help the reader identifying the cells, red arrows were added to the fluorescence acquisition.

As already seen in MTT assays, the functionalization with FBS improves adhesion capability of cells, which is suggested by looking at the cell morphology (Figure 10) that is the same of PC-3 cultured on standard substrates.

#### 3.2.3. Cell Migration Proof-of-Concept

The potential of microfabricated membranes obtained by femtosecond laser to be used as a tool for biological investigations has been evaluated by setting up a migration assay on multilayered CKC membranes. These were selected because of pores dimensions which were more suitable according to cells dimensions.

After FBS functionalization, gold-coated CKC membranes were put in 35 mm petri dishes. Membranes were suspended from the bottom by a PDMS ring. PC3 cells were then seeded on the tapered surface of CKC membranes (seeding side) (Figure 10a) and allowed to adhere in standard cell culturing conditions for 24 h. At the end of the experiment, the set up was disassembled and the membrane was observed on the reverse side (migration side). As can be seen in Figure 10b, cells were able to migrate through the pores and to reach the reverse surface.

## 4. Conclusions

In this paper, we have proved the feasibility of fabricating membranes with controlled micropores distribution by femtosecond laser ablation. With a single laser source and experimental setup, membranes of two different materials, i.e., CKC multilayer and CA, were fabricated, but it can be applicable to any other material.

In particular, large area distribution of pores with a precise control of the micrometric pore distribution has been demonstrated. A proper selection of the irradiation parameters allowed producing pores with dimension of 8 µm and 18 µm on CKC and CA substrates, respectively.

On average, the fabrication process requires few tens of minutes per filter, and it is up to 100 times more efficient, intended as time per hole, than in previous laser-based membrane fabrication work [29]. However, in order to furtherly reduce this time and produce membranes on mass scale, the next step of this study could be exploiting, e.g., a multiple-beam scanning method using a spatial light modulator (SLM) [40].

For evaluating the possibility to use such laser-fabricated membranes into biological assays, their biocompatibility and suitability for migration assays have been investigated. To this aim, as a proof of concept, we tested the two materials into viability tests. The culture of mammalian cells on these microfabricated membranes were studied showing that CKC and CA membrane are suitable for biological investigations, e.g., for in-vitro modelling of physiological barriers (e.g., lungs, intestinal epithelium) and co-culture studies. This membrane can be functionalized with different extracellular matrix protein to perform migration assays and evaluate the effect of environmental composition of cell metastatic potential. In addition, the CKC membranes having a conductive copper layer can be used for monitoring the cells movements through electrical signals.

## Figures and Tables

**Figure 1 micromachines-13-01371-f001:**
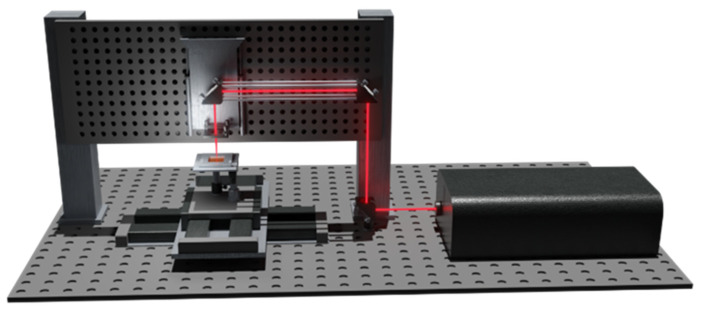
Sketch of the laser-based microfabrication setup. The laser source was carried and focalized over the sample, stuck on the translational stage.

**Figure 2 micromachines-13-01371-f002:**
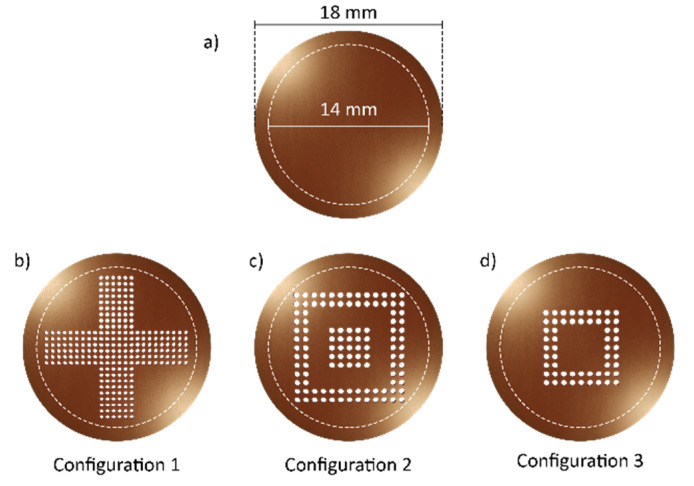
Design of the micromembrane. The main dimensions are indicated in (**a**–**d**) report on the different pores’ distribution in the three configurations. Pores’ size not to scale.

**Figure 3 micromachines-13-01371-f003:**
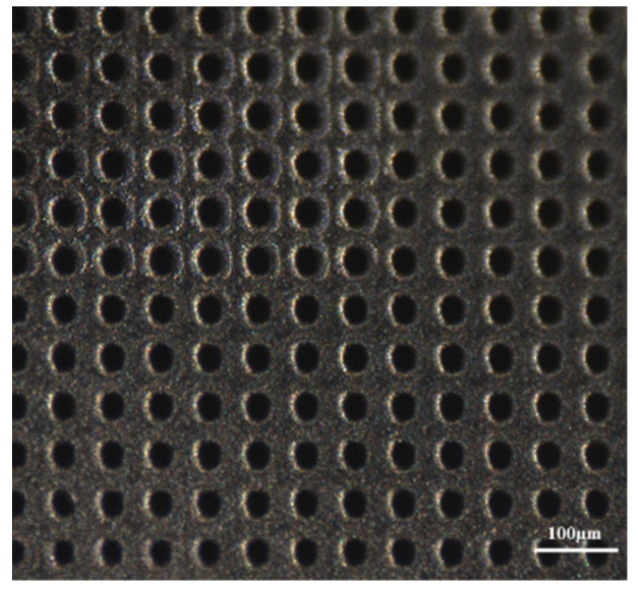
Gold-coated CKC membrane before being functionalized and used for viability tests (migration side).

**Figure 4 micromachines-13-01371-f004:**
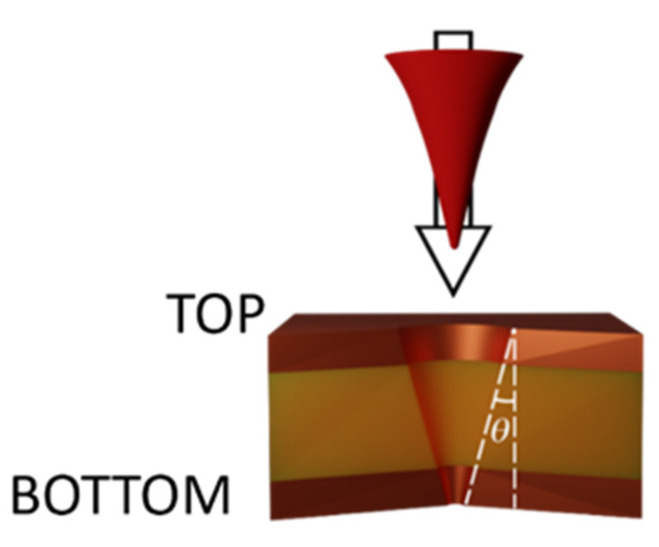
Pore cross section. θ represents the taper angle.

**Figure 5 micromachines-13-01371-f005:**
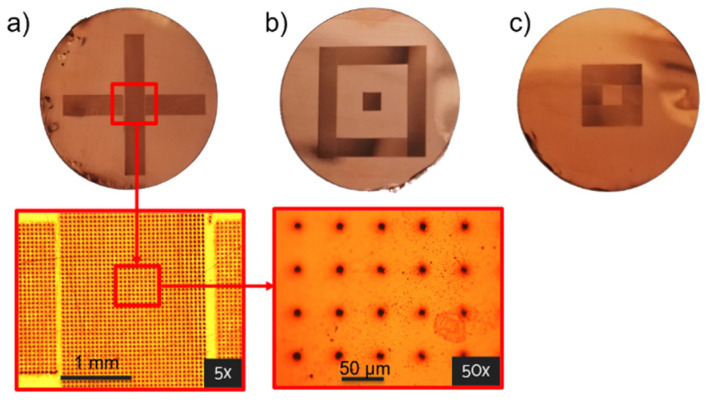
Pictures of the three layouts (Configuration 1, 2 and 3 respectively in section (**a**–**c**) of the figure) reproduced by fs-laser drilling on the CKC bottom surface. Pore dimensions: 8 ± 1 µm. Distance between pores: 50 µm.

**Figure 6 micromachines-13-01371-f006:**
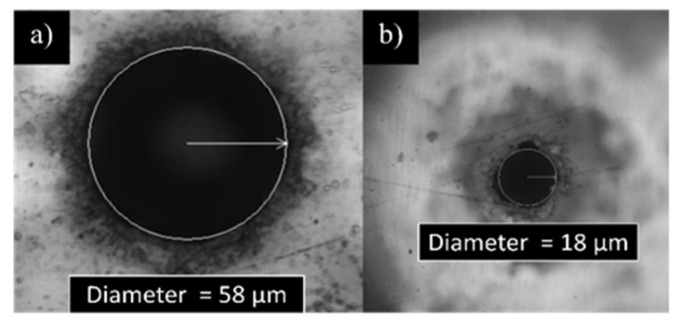
(**a**) Top and (**b**) bottom side of a fs-laser drilled pore on CA layer. The average dimensions are indicated.

**Figure 7 micromachines-13-01371-f007:**
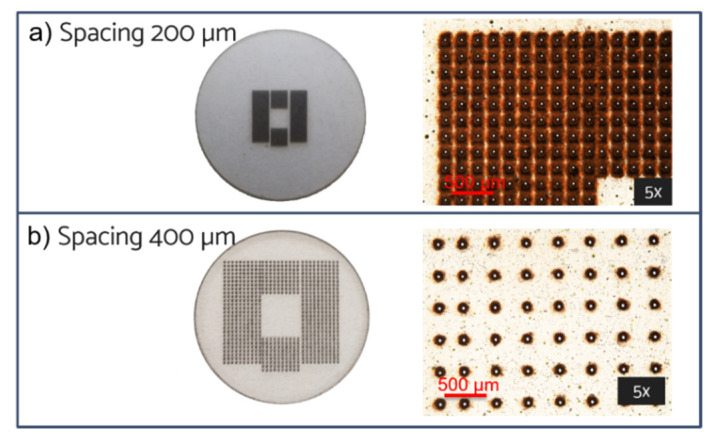
Configuration 3 on CA bottom surface. Distance between micropores (**a**) 200 µm and (**b**) 400 µm. Pore exit dimensions: 18 ± 2 μm.

**Figure 8 micromachines-13-01371-f008:**
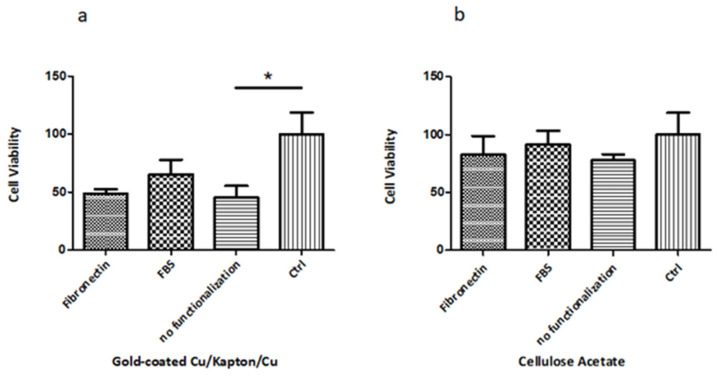
MTT Assay of PC-3 cells in culture medium conditioned with (**a**) CKC fragments coated with gold monolayer and (**b**) CA fragments functionalized with Fibronectin, FBS and no functionalization than control.

**Figure 9 micromachines-13-01371-f009:**
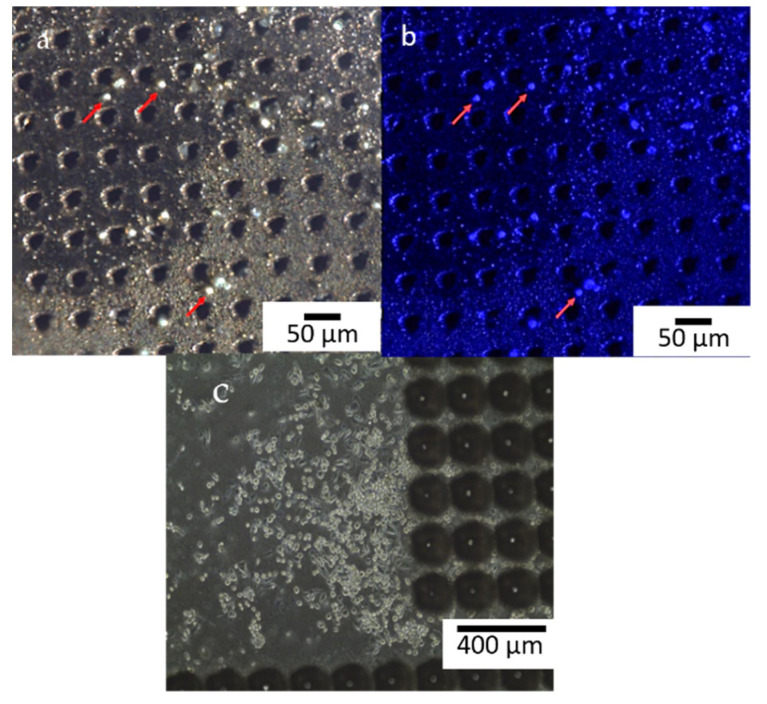
(**a**) Brightfield image of PC-3 cells seeded on multilayer CKC pores membrane coated with a nanometric gold layer to make it biocompatible; (**b**) Fluorescence image of PC-3 nuclei stained with DAPI between the pores of the Gold-CKC membrane. Red arrows identify (**c**) PC-3 cells seeded on CA membrane.

**Figure 10 micromachines-13-01371-f010:**
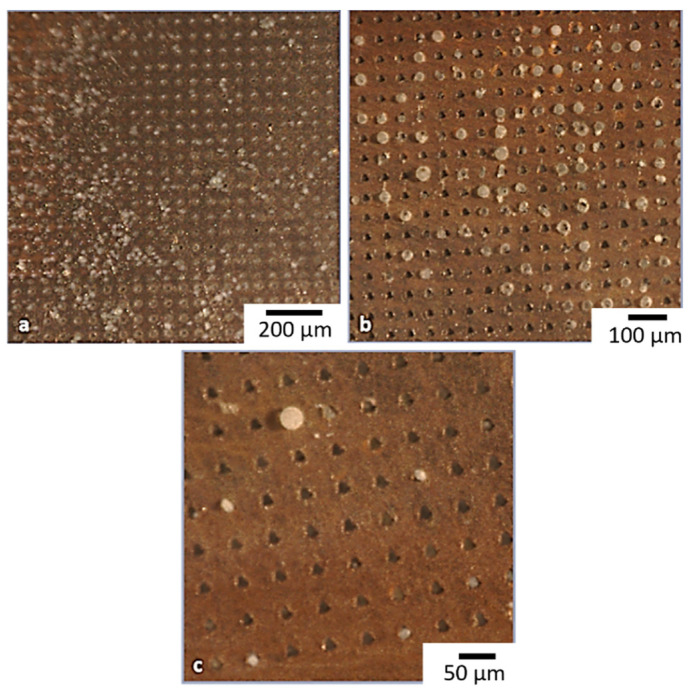
(**a**) Gold-coated CKC membranes used for cell migration experiments after functionalization with FBS. Cells spread over the whole surface on the seeding side. (**b**) and magnification in (**c**) 24 h after seeding, the membrane was observed on the reverse side (migration side). As can be seen cells sprout out from the pores thus demonstrating that membranes can be used in migration assays.

**Table 1 micromachines-13-01371-t001:** Laser working parameters.

	Cu-Kapton-Cu (CKC)	CA
Wavelength (nm)	1030	1030
Pulse duration (fs)	200	200
Power (mW)	250	30
Repetition Rate (kHz)	50	0.05
Pulse Energy (µJ)	5	1 10^3^
Fluence	22.6	45.3 × 10^2^
Shots per hole	2500	30

**Table 2 micromachines-13-01371-t002:** Characteristics of CKC membrane configurations. The values are referred to the bottom surface of the membranes. Configurations 1, 2 and 3 are referred to as the description in Figure 5a–c.

	Configuration 1	Configuration 2	Configuration 3
Pores number	15,925	20,825	9800
Pore-pore distance (µm)	50	50	50
Pore exit diameter (µm)	8 ± 1	8 ± 1	8 ± 1
Pore density (pores/mm^2^)	103	135	63
Fabrication time (min)	40	52	25

**Table 3 micromachines-13-01371-t003:** Characteristics of CA membrane configurations. The values are referred to the bottom surface of the membranes.

	Configuration 1	Configuration 2	Configuration 3
Pores number	4440	5806	2732
Pore-pore distance (µm)	200–400	200–400	200–400
Pore exit diameter (µm)	18 ± 2	18 ± 2	18 ± 2
Pore density (pores/mm^2^)	30	38	18
Fabrication time (min)	81	106	50

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
