# Peer review of "Femtosecond Laser Fabrication of Microporous Membranes for Biological Applications"

_micromachines, 2022, doi:10.3390/mi13091371_

Round 1

Reviewer 1 Report

In this manuscript, the authors claimed the fabrication of microporous membranes by femtosecond laser on two materials: CKC and CA. Also, they evaluated the possibility of the fabricated membranes in biological assay. However, femtosecond laser drilling of metals, polymers, semiconductors, and ceramics have been widely investigated, including improving fabrication precision, the perpendicularity of the hole and efficiency, et al. What’s the novelty of this manuscript? The reason of the choice of these two materials is not clear. CKC is not the commonly used biological material. Therefore, according to the description and experimental data presented in the manuscript, I don’t think it can be published in micromachines.

(1) The innovation of the manuscript is not clear. Why they choose these two materials? What’s the difficulty of femtosecond laser drilling of these two materials compared with other materials, for example copper (DOI10.1002/adem.202000710), PMMA (DOI10.1063/1.4981248).

(2) the authors claimed that “they obtained microfilters with pore size from 8 to 20 µm in diameter, high aspect ratio (the ratio of the thickness of the material over the diameter of the hole is up to 8:1)” However, they only presented pore size of 8 µm on CKC and 18 µm on CA. I just wondered if pore size of 8 µm can be realized on CA? What’s more, the calculation of aspect ratio is inaccuracy as the diameter of the inlet and outlet side of the the hole is different. For a conical hole, conicity can accurately describe the morphology of the hole.

(3) figures 9 and 10 are not clear. The text in Figure 9 (a) and (b), and Figure 10 can’t be recognized.

Author Response

We have revised the paper according to the referee’s suggestions. In the following I have enclosed a detailed response to all of the reviewer’s points. All the revisions to the manuscript have been marked up using the “Track Changes” function of MS Word. The lines number indicated in the following are referred to the text without track changes highlighted.

I hope the paper may be now accepted for publication in Micromachines, MPDI.

Yours sincerely,

Annalisa Volpe

---------

In this manuscript, the authors claimed the fabrication of microporous membranes by femtosecond laser on two materials: CKC and CA. Also, they evaluated the possibility of the fabricated membranes in biological assay. However, femtosecond laser drilling of metals, polymers, semiconductors, and ceramics have been widely investigated, including improving fabrication precision, the perpendicularity of the hole and efficiency, et al. What’s the novelty of this manuscript? The reason of the choice of these two materials is not clear. CKC is not the commonly used biological material. Therefore, according to the description and experimental data presented in the manuscript, I don’t think it can be published in micromachines.

The authors thank the referee for considering our work. In the following, we provide explanations for the points raised by him/her, hoping to make the purpose of our work clearer.

  • The innovation of the manuscript is not clear. Why they choose these two materials? What’s the difficulty of femtosecond laser drilling of these two materials compared with other materials, for example copper (DOI10.1002/adem.202000710), PMMA (DOI10.1063/1.4981248).

Re: We agree with the reviewer that laser drilling of different materials has been widely investigated, Therefore, in this work we do not provide new insights in the laser drilling process itself. The scope of this paper is to show the potential of femtosecond laser drilling to fabricate, in a single step, membranes for microfluidic biomedical devices, with high precision, reproducibility and great flexibility since this technology offers the possibility to tune very easily the geometrical arrangement and size of the pores and there is almost no restriction on the type of substrate material chosen for each specific application (see abstract, lines 18-19 and introduction 130-137). Currently, as argued in the introduction, for the fabrication of membranes a plethora of techniques are used depending on the material. Many of these techniques require complex facilities and are non-reconfigurable. Following the reviewers’ suggestions, we have more clearly specified our aim and stated the novelty of our work in lines 130-138 of the manuscript.

As regards the choice of the materials, since the laser technique can potentially be used on any substrate, as a proof of concept, we have chosen two very different ones. CA, a transparent polymer, widely used for biological applications and highly biocompatible, and CKC, an opaque conductive material. However, these two do not exhaust the plethora of materials that can be machined, just by adjusting the laser parameters. At the state of the art, CKC is not commonly used in biological essays. However, being composed of two layers of conductive material, isolated by a dielectric layer, we believe that CKC has a great potential for being used for the monitoring of the passage of cells through the membranes that could be detected by an electric signal if the conductive layers are charged. We outline this aspect in section 3.1.

  • the authors claimed that “they obtained microfilters with pore size from 8 to 20 µm in diameter, high aspect ratio (the ratio of the thickness of the material over the diameter of the hole is up to 8:1)” However, they only presented pore size of 8 µm on CKC and 18 µm on CA. I just wondered if pore size of 8 µm can be realized on CA? What’s more, the calculation of aspect ratio is inaccuracy as the diameter of the inlet and outlet side of the the hole is different. For a conical hole, conicity can accurately describe the morphology of the hole.

Re: The cited sentence (lines 20-21) has been modified in order to make clearer that the pore sizes of 8 and 20 um have been obtained on CKC and CA, respectively. These values are the minimum values that we were able to obtain with the exploited setup. However, we do not exclude to be able to do better using, eg., a different microscope objective.

We agree with the referee that in case of conical hole, calculating the aspect ratio, considering the smallest diameter, and neglecting the largest one, is not accurate. For this reason, we have removed any reference to the aspect ratio, just describing the taper of the hole.

  • figures 9 and 10 are not clear. The text in Figure 9 (a) and (b), and Figure 10 can’t be recognized.

Re: We thank the referee for helping us improve the work. We have made figures 9 and 10 more legible highlighting the scale bars.

Reviewer 2 Report

The authors report on the femtosecond laser 18 fabrication of micromembranes both on polymeric and metallic substrate for application in cell migration studies.

The paper describes the method, setup and application of the technique. Some points need to be clarified before publication.

1.     A clear characterization of fabricated membrane is missed. The CKC membrane shown in figure 4 has pore dimension of 8 microns and period of 50 microns. However, figure 5 shown a conical shape of the pores. Does the 8 microns refer to the larger or smaller aperture?

2.     Table 2 reports Title 1, 2 and 3, but only two types of membrane are described. The author should change the first raw as in table 3

3.     Figure 2 shows gold coated CKC membrane with pores of 20-30 microns diameter instead of the 8 microns of that in figure 4? Is it the same CKC membrane of figure 4? Is it the same fabrication method?

4.     Since the membrane is thicker with respect the Rayleigh range of the gaussian beam a detailed characterization of the focused beam must be reported. The author just say that the calculated beam is 4.7 micron. How was this value obtained? What is the Rayleigh range of the Gaussian beam?

5.     How do the authors explain the larger pore size (8 microns) with respect to the 4.7 calculated beam waist?

6.     What is the smallest size of the pore dimension? Does it correspond to the calculated 4.7 micron gaussian beam waist? How does the pore dimension change with the number of laser shots?

7.     During the fabrication of the single pore, was the gaussian beam focused onto the first surface, onto the second or in the middle of the membrane?

8.     What do the author mean when they stated that this method is “…up to 100 times more efficient than in previous laser-based membrane fabrication work..” (line 344)?

Author Response

We have revised the paper according to the referee’s suggestions. In the following I have enclosed a detailed response to all of the reviewer’s points. All the revisions to the manuscript have been marked up using the “Track Changes” function of MS Word. The lines number indicated in the following are referred to the text without track changes highlighted.

I hope the paper may be now accepted for publication in Micromachines, MPDI.

Yours sincerely,

Annalisa Volpe

---------

The authors report on the femtosecond laser 18 fabrication of micromembranes both on polymeric and metallic substrate for application in cell migration studies.

The paper describes the method, setup and application of the technique. Some points need to be clarified before publication.

The authors would like to thank the reviewer to carefully read our work. We have taken advantage of the useful suggestions proposed to improve the manuscript, as better described below.

  1. A clear characterization of fabricated membrane is missed. The CKC membrane shown in figure 4 has pore dimension of 8 microns and period of 50 microns. However, figure 5 shown a conical shape of the pores. Does the 8 microns refer to the larger or smaller aperture?

Re: The make clearer the characterization of the holes, section 3.1.1 has been rearranged, the top and bottom surface have been labelled in fig 4. According to these definitions, the captions of tab2, tab3, fig 5 and fig 7 have been changed.

  1. Table 2 reports Title 1, 2 and 3, but only two types of membrane are described. The author should change the first raw as in table 3

Re: The authors thank the reviewer for pointing out the oversight. In tab 2, the titles have been changed and the parameters relating to configuration 3 has been added, too.

  1. Figure 2 shows gold coated CKC membrane with pores of 20-30 microns diameter instead of the 8 microns of that in figure 4? Is it the same CKC membrane of figure 4? Is it the same fabrication method?

Re: Figure 2 is referred to the top side (migration side) of the membrane. Conversely, fig 4 (now 5) is referred to the other side (seeding side), as now described in the captions and in the main text.

  1. Since the membrane is thicker with respect the Rayleigh range of the gaussian beam a detailed characterization of the focused beam must be reported. The author just say that the calculated beam is 4.7 micron. How was this value obtained? What is the Rayleigh range of the Gaussian beam?

Re: The description of the laser beam (section 2.2) has been modified, inserting the formula to calculate the beam minimum diameter and taking into account the value of M2 (∼1.3). The Rayleigh range has been inserted, too.

  1. How do the authors explain the larger pore size (8 microns) with respect to the 4.7 calculated beam waist?

Re: The calculated value reported implies a perfect location of the focal position on the sample surface. In our case, the using a microscope objective with a short depth of focus makes this particularly complicated, in particular considering that the sample could not be perfectly flat. For these reasons, we set the laser parameters in order to be sure to generate holes on all the sample even if not as small as possible.

  1. What is the smallest size of the pore dimension? Does it correspond to the calculated 4.7 micron gaussian beam waist? How does the pore dimension change with the number of laser shots?

Re: In the preliminary study, the minimum pore dimensions (bottom side) obtained both for CKC and CA was approximately 5 microns. However, this value was found just for a few holes in a few tens of , after a precise search for the focus.

  1. During the fabrication of the single pore, was the gaussian beam focused onto the first surface, onto the second or in the middle of the membrane?

Re: The non-perfect flatness of the substrates does not allow to have the certainty of the precise focal position during the entire laser processing, in particular if, as in this case, some cm2 of area must be machined.

  1. What do the author mean when they stated that this method is “…up to 100 times more efficient than in previous laser-based membrane fabrication work..” (line 344)?

Re: Here, the efficiency is intended as machining time per hole, as now also clarified in the text (line 354).

Reviewer 3 Report

The work applied a femtosecond laser to produce microporous membranes on CKC and CA for biological applications. The idea is helpful for practical uses. However, the obtained results have not been discussed well. Several statements should be placed in the experimental rather than the result.  Therefore, the authors should improve the manuscript quality to be qualified to publish in the Micromachines.

1)     Figure 3 and its description should belong to part 2.2 Micro-membrane design and fabrication

2)     In Table 2, what do titles 1, 2, and 3 represent?

3)     What is the taper angle in the case of CA?

4)     Are there any references or analyzed data to support this statement “This difference can be attributed to a combined effect of the different substrate thickness and to the base material.”

5)     What is the mechanism/reason for more pronounced cell viability on the CA membrane than on the CKC one?

6)     In Figure 9, please describe the red arrows.

7)   The statement, “PC3 cells were then seeded on the tapered surface of CKC membranes (seeding side) (Figure 10 a) and allowed to adhere in standard cell culturing conditions for 24 hours. At the end of the experiment,  the set up was disassembled and the membrane was observed on the reverse side (migration side)”, should be described in the experimental part.

8)     The texts on some figures are hard to see.

9)     Several grammatical mistakes should be corrected.

Author Response

We have revised the paper according to the referee’s suggestions. In the following I have enclosed a detailed response to all of the reviewer’s points. All the revisions to the manuscript have been marked up using the “Track Changes” function of MS Word. The lines number indicated in the following are referred to the text without track changes highlighted.

I hope the paper may be now accepted for publication in Micromachines, MPDI.

Yours sincerely,

Annalisa Volpe

-------

The work applied a femtosecond laser to produce microporous membranes on CKC and CA for biological applications. The idea is helpful for practical uses. However, the obtained results have not been discussed well. Several statements should be placed in the experimental rather than the result.  Therefore, the authors should improve the manuscript quality to be qualified to publish in the Micromachines.

The authors would like to thank the reviewer for having considered the idea helpful. We have taken advantage of the useful suggestions proposed to improve the manuscript, as better described below.

Figure 3 and its description should belong to part 2.2 Micro-membrane design and fabrication

Re: According to the reviewer suggestion, Fig 3 and its description have been moved to section 2.2. Consequently, section 3.1 has been completely removed.

  • In Table 2, what do titles 1, 2, and 3 represent?

Re: The authors thank the reviewer for pointing out the oversight. In tab 2, the titles have been changed and the parameters relating to configuration 3 has been added, too.

  • What is the taper angle in the case of CA?

Re: In the case of CA, the taper angle is of about 10°, as well as in the case of CKC. This info has been added in section 3.1.2 (line 261).

  • Are there any references or analyzed data to support this statement “This difference can be attributed to a combined effect of the different substrate thickness and to the base material.”

Re: The sentence has been rephrased. “This difference can be attributed to the base material which, being a dielectric, dissipates heat worse than copper causing unwanted melting at the edges of the hole.”

5)    What is the mechanism/reason for more pronounced cell viability on the CA membrane than on the CKC one?

Re: CKC membrane consists of a multi-layered structure in which a Kapton layer is enclosed between two layers of Copper. Copper is known to be very toxic for cells already at low micromolar concentrations, as demonstrated by a high number of published papers. We have now added some of them to the revised version of the manuscript in section 2.3 (line 200). So, in order to make the CKC membranes biocompatible, we deposited a gold coating on both faces of them, thus covering the most of the surface with a cell-friendly metal. Anyway, by using a thermal evaporation method, the coating cannot be total and some areas of the membrane could not be completely isolated (thickness of CKC substrate in the perimeter and into the pores) thus generating a slight cytotoxic effect. CA indeed has established biocompatibility features.

6)     In Figure 9, please describe the red arrows.

Re: The arrows we inserted in Figure 9 are intended to facilitate the identification of cells in the brightfield and  fluorescence microscope image. We added this instruction in the manuscript text and in the caption of figure 9.

7)   The statement, “PC3 cells were then seeded on the tapered surface of CKC membranes (seeding side) (Figure 10 a) and allowed to adhere in standard cell culturing conditions for 24 hours. At the end of the experiment,  the set up was disassembled and the membrane was observed on the reverse side (migration side)”, should be described in the experimental part.

Re: We thank Reviewer 2 for this suggestion. We have now added this experimental description also in Materials and methods section (Section 2.3).

8)     The texts on some figures are hard to see.

Re: We thank the referee for helping us improve the work. We have made figures 9 and 10 more legible highlighting the scale bars.

9)     Several grammatical mistakes should be corrected.

Re: The text has been revised and the grammatical mistakes have been corrected.

Round 2

Reviewer 1 Report

Although the authors made enough responses to the comments, I still can’t find the novelty of the work. Anyway, they have made some efforts to evaluate the possibility to use such laser-fabricated membranes into biological assay. Some points can be considered.

1. some discussions should be provided about the appropriate size of the hole needed for cells proliferation or adhesion. From this point of view. The authors may not need to make very small holes (8 um in the paper).

2. the figures 9 and 10 still doesn't look very clear (it’s too dark, the details of the morphology of cells can’t be seen), not just the text. If possible, a high magnification image may be more suitable.

Author Response

Reviewer 1 (round 2)

Although the authors made enough responses to the comments, I still can’t find the novelty of the work. Anyway, they have made some efforts to evaluate the possibility to use such laser-fabricated membranes into biological assay. Some points can be considered.

We thank the referee for considering the efforts made to improve our work. In the following, we have discussed the points to be considered.

  1. some discussions should be provided about the appropriate size of the hole needed for cells proliferation or adhesion. From this point of view. The authors may not need to make very small holes (8 um in the paper).

Re: We thank the reviewer for this question. What we have tried to replicate into our work, is a migration assay, which is usually performed on polycarbonate membranes in what is called “Boyden Chamber”. In a standard Boyden assay, the pore diameter of the membrane is typically 3 to 12 μm, and it is selected to suit the subject cells (https://doi.org/10.3389/fcell.2019.00107 ). We fixed 8 μm because we think it is a good average, useful for biological assays and it is a good compromise to allow migration assay but also to permit cells to adhere and proliferate on a regular surface. Pores too big on the membrane won’t represent a suitable adhesion surface for cells used to grow as a monolayer. Moreover, migration assays usually make use of pores with diameters very lower with respect to cell diameter because cells able to invade tissues from blood (i.e., circulating tumour cells or methastatic cells) show an enhanced flexibility. This makes them able to cross little endothelial pores by exerting mechanical changes which are shown to induce transition to an invasive phenotype. DOI: 10.1038/s41567-019-0680-8.

  1. the figures 9 and 10 still doesn't look very clear (it’s too dark, the details of the morphology of cells can’t be seen), not just the text. If possible, a high magnification image may be more suitable.

Re: Fig 9 and 10 have been modified in order to make the details clearer. Being the surface not transparent (in the case of CKC substrate) we had to apply epiluminescence to perform the experiments, that’s why the appearance of the images is quite dark.

Reviewer 2 Report

The authors have addressed the revisions requested.

Author Response

We thank the referee again for the useful advice and for considering our manuscript suitable for publication on Micromachines.

Reviewer 3 Report

The authors have improved their work quality. Therefore, this research work is suggested to be qualified for publication in Micromachines

Author Response

we thank the referee again for the useful advice and for considering our manuscript suitable for publication on Micromachines

Round 3

Reviewer 1 Report

The manuscript can be accepted.